

# Finding priority bacterial ribosomes for future structural and antimicrobial research based upon global RNA and protein sequence analysis

Helena B. Cooper[1,2], Kurt L. Krause[1] and Paul P. Gardner[1]

[1] Department of Biochemistry, University of Otago, Dunedin, New Zealand
[2] Department of Infectious Diseases, Central Clinical School, Monash University, Melbourne, Victoria, Australia

Corresponding author
Paul P. Gardner,
paul.gardner@otago.ac.nz

## ABSTRACT

Ribosome-targeting antibiotics comprise over half of antibiotics used in medicine, but our fundamental knowledge of their binding sites is derived primarily from ribosome structures of non-pathogenic species. These include *Thermus thermophilus*, *Deinococcus radiodurans* and the archaean *Haloarcula marismortui*, as well as the commensal and sometimes pathogenic organism, *Escherichia coli*. Advancements in electron cryomicroscopy have allowed for the determination of more ribosome structures from pathogenic bacteria, with each study highlighting species-specific differences that had not been observed in the non-pathogenic structures. These observed differences suggest that more novel ribosome structures, particularly from pathogens, are required for a more accurate understanding of the level of diversity of the entire bacterial ribosome, with the potential of leading to innovative advancements in antibiotic research. In this study, high accuracy covariance and hidden Markov models were used to annotate ribosomal RNA and protein sequences respectively from genomic sequence, allowing us to determine the underlying ribosomal sequence diversity using phylogenetic methods. This analysis provided evidence that the current non-pathogenic ribosome structures are not sufficient representatives of some pathogenic bacteria, such as *Campylobacter pylori*, or of whole phyla such as Bacteroidota (Bacteroidetes).

## SIGNIFICANCE STATEMENT

The growing number of antibiotic resistance pathogenic bacteria are of critical concern to the health profession. Many of the current classes of antibiotics target the bacterial ribosome, the protein making factory for these species. However, much of our knowledge of the bacterial ribosome is based upon non-pathogenic bacteria that are highly divergent from the major pathogens of concern. We have analysed the genetic variation of the RNA and protein components of all available bacterial ribosomes. This has led us to identify the highest priority groups of bacteria that would provide the most benefit most from further analysis of their ribosome structures, from both a medical and evolutionary perspective.
## INTRODUCTION

A global rise of antibiotic resistant bacteria has become an increasingly urgent problem in recent years, with the bacterial ribosome, particularly the ribosomal RNA (rRNA) component, being a common antibiotic target (*Tacconelli et al., 2018*; *Lin et al., 2018*; *Wilson, 2014*). Due to the limitations and requirements of X-ray crystallography, antibiotic binding studies initially used extremophiles *Thermus thermophilus* (*Polikanov et al., 2015*), *Deinococcus radiodurans* (a 50S structure) (*Harms et al., 2008*; *Kaminishi et al., 2015*) and the archaeon *Haloarcula marismortui* (*Ban et al., 2000*), with the pathogenic *Escherichia coli* (*Fischer et al., 2015*; *Noeske et al., 2015*; *Watson et al., 2020*) ribosome structure introduced in later years (*Lin et al., 2018*; *Wilson, 2014*; *Fox, 2010*). However, improvements in electron cryomicroscopy have allowed more diverse ribosome structures to be analysed more recently, such as from non-pathogenic bacteria *Bacillus subtilis* (*Sohmen et al., 2015*; *Crowe-McAuliffe et al., 2021a*), *Flavobacterium johnsoniae* (*Jha et al., 2021*) and *Mycobacterium smegmatis* (*Hentschel et al., 2017*) (recently renamed "*Mycolicibacterium smegmatis*"), or from pathogens including *Acinetobacter baumannii* (*Morgan et al., 2020*; *Zhang et al., 2021*), *Enterococcus faecalis* (*Murphy et al., 2020*; *Crowe-McAuliffe et al., 2021b*), *Listeria monocytogenes* (*Crowe-McAuliffe et al., 2021b*; *Koller et al., 2022*), *Mycobacterium tuberculosis* (a 50S structure) (*Yang et al., 2017*), *Pseudomonas aeruginosa* (a 50S structure) (*Halfon et al., 2019*) and *Staphylococcus aureus* (*Crowe-McAuliffe et al., 2021b*; *Eyal et al., 2015*; *Halfon et al., 2019*). Each pathogen ribosome study highlights species-specific differences in comparison to both non-pathogenic and pathogenic structures, implying that non-pathogenic ribosome structures are not always optimal for inferring antibiotic binding across all bacteria (*Morgan et al., 2020*; *Yang et al., 2017*; *Halfon et al., 2019*; *Eyal et al., 2015*). As these structural differences could hinder antibiotic research it is important that solved ribosome structures, which form the basis of our understanding of the bacterial ribosome and ribosomal antibiotic binding, are suitably representative of pathogenic species (*Lin et al., 2018*; *Wilson, 2014*).

We further note that the specific target sites of antibiotics correspond to highly conserved, functional ribosome components, and that a survey focusing on these sites will not yield significant diversity. However, we also note that ribosome function has been found to be influenced by less well conserved components of structure (*Lin et al., 2018*; *Eyal et al., 2015*), and even by elements of structure quite distant from conserved antibiotic binding sites (*Belousoff et al., 2017*). In addition, there is evidence in *S. aureus* and *P. aeruginosa* suggesting that changes to the overall ribosome can lead to antibiotic resistance, in addition to direct mutations in the target sites (*Halfon et al., 2019*; *Belousoff et al., 2017*).

Therefore, the aim of this study is to broadly survey ribosome sequence space to determine how representative current ribosome structures are, ranging from the extremophiles (*e.g.*, *D. radiodurans*, *T. thermophilus*) for human pathogenic species (*e.g.*, *S. aureus*, *L. monocytogenes*), when compared to all bacterial species. Using this analysis, we propose new representative bacteria, including some surprising and diverse species,

with the potential of yielding valuable new ribosome structural information that could be prioritised for future ribosomal structural studies.

## MATERIALS AND METHODS

Our analysis includes the full-length conserved components of the bacterial ribosome. We have elected not to specifically analyse antibiotic targets (*e.g.*, Spectinomycin targets G1064 and C1992 of 16S rRNA (*Brink et al., 1994*)) as these are highly conserved functional components of the ribosome, which therefore carry a limited number of phylogenetically informative characters (*Petrov et al., 2014*; *Bernier et al., 2018*; *Tirumalai et al., 2021*). These studies also illustrate the conservation and variation across ribosomal RNA structures. Full-length sequences also allow a global overview of ribosome variation, some of which may change the behaviour of binding sites due to "action at a distance" (*Lin et al., 2018*; *Eyal et al., 2015*; *Belousoff et al., 2017*).

A total of 3,758 bacterial genomes and the *H. marismortui* genome (Accession: AY596297.1) were obtained from the European Nucleotide Archive (*Amid et al., 2020*; *Baliga et al., 2004*). One representative sequence was retained per species for each rRNA and ribosomal protein sequence, which were filtered to only include genomes with annotations covering 80% of the expected sequence length based upon consensus sequences. If paralogues were present, the sequence with the highest corresponding covariance model and hidden Markov model bit score for the species was used (*Nawrocki & Eddy, 2013*).

Both 16S and 23S rRNA were annotated using barrnap v0.9 (https://github.com/tseemann/barrnap), and INFERNAL 1.1.2 was used to create sequence alignments using Rfam v14.3 covariance models (*Nawrocki & Eddy, 2013*; *Kalvari et al., 2018*). Covariance models are the gold-standard for structural RNA alignment (*Nawrocki & Eddy, 2013*; *Freyhult, Bollback & Gardner, 2007*), and the Rfam models are derived from the CRW database (*Cannone et al., 2002*), which is the product of 20 years of manual alignment curation, resulting in some of the highest quality ribosomal RNA structural alignments available to date (*Cannone et al., 2002*; *Gutell, Lee & Cannone, 2002*).

Ribosomal protein sequences from 32 universally conserved proteins were identified from six-frame translations of whole genomes, to account for potentially inconsistent or absent genome annotations. The selected sequences were aligned using HMMER v3.1 (hmmer.org), with protein hidden Markov models (HMMs) from Pfam v33.1 (*El-Gebali et al., 2019*). Profile HMMs are the gold-standard for protein homology search and alignment (*Eddy, 2011*; *Söding & Remmert, 2011*), while Pfam is a widely used and highly curated database of protein domain alignments.

The advantages of this approach are that our results are up-to-date and consistent across the current genome databases, contain both RNA and protein components of the bacterial ribosome and utilise the most accurate methods for alignment production. Whilst we acknowledge that there are existing ribosomal datasets (*e.g.*, SILVA, GreenGenes, RDB, *etc.*), these do not meet all the above requirements. Although there are alternative ribosomal RNA and protein alignments we could have used (*e.g.*, *Bernier et al., 2018*; *Doris et al., 2015*; *Hug et al., 2016*), their use here would likely produce very similar results.

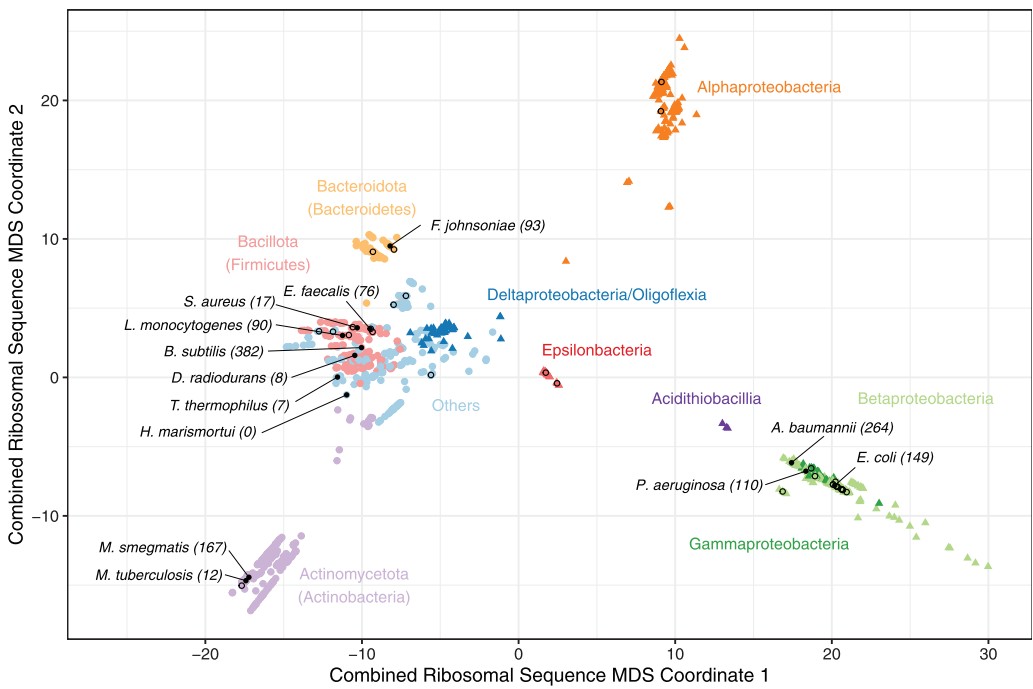

**Figure 1 A multiple-dimensional scaling (MDS) plot for combined phylogenetic distances from 16S, 23S rRNA and 26 universally conserved ribosomal proteins.** A multiple-dimensional scaling (MDS) plot for combined phylogenetic distances from 16S, 23S rRNA and 26 universally conserved ribosomal proteins ($N$ = 1,396). All Pseudomonadota (Proteobacteria) are coloured by class (triangles), and other included phyla (circles) are coloured individually to highlight taxonomic relationships with clusters. Species with a solved ribosome structure ($N$ = 13), or known pathogenic bacteria without a structure ($N$ = 38) have been highlighted with a filled or hollow black dot respectively. The number of bacteria that consider each species with solved structure to be representative, based on the minimum phylogenetic distance, has been labelled along with the species' names. The full list of species, MDS coordinates, the solved structures that were considered to be most representative and the corresponding minimum distance are available on GitHub (https://github.com/helena-bethany/ribosomal-variation).

Phylogenetic trees for each alignment were generated using the maximum likelihood method from PHYLIP v3.697, with distance matrices of the pairwise distances between species computed in R v4.2.1 using ape v5.6 (*Felsenstein, 2009*; *Paradis & Schliep, 2019*). The distance matrices for each ribosomal gene were summed to create a single unified distance matrix, and this was used for multi-dimensional scaling (MDS) and visualisation (Fig. 1).

## RESULTS

Ribosomal sequence clustering suggests that several bacterial phyla are poorly represented by the currently available ribosome structures. To identify phyla that are poorly represented by solved ribosome structures, two rRNA and 32 universally conserved ribosomal protein phylogeny trees were initially created using a representative sequence for each species. To confirm that these universally conserved ribosomal proteins were truly conserved in our dataset, the trees were filtered on whether they contained more than 95% of isolates from each taxonomy class in the dataset. Taxonomy class was used over phyla to

take into account variability within larger, overrepresented phyla such as Pseudomonadota (Proteobacteria) (*Hug et al., 2016*). This resulted in ribosomal proteins uS2, uS4, uL23, uL24 and uL30 being removed because they were completely absent in Epsilonproteobacteria and uL15 being removed as it was only present in 16% of Epsilonproteobacteria genomes. As assessing individual ribosomal sequences is unlikely to reflect structural variation, the remaining two rRNA and 26 ribosomal protein phylogenetic trees were combined into one distance matrix, resulting in a final dataset of 1,396 species. This combined tree was then reduced using MDS, allowing us to visualise the most dissimilarities between species (*Hug et al., 2016*).

The resulting MDS plot shows four main clusters; an Alphaproteobacteria cluster and a Betaproteobacteria plus Gammaproteobacteria cluster illustrate the non-monophyletic nature of Pseudomonadota (Proteobacteria) (*Hug et al., 2016*), one is dominated by Actinomycetota (Actinobacteria) and the fourth consists of the remaining bacterial phyla and Proteobacterial classes (Fig. 1). The Pseudomonadota (Proteobacteria) cluster follows observations made in previous studies, such as Beta and Gammaproteobacteria being closely related, and Deltaproteobacteria and Oligoflexia being more related to non-Proteobacteria (*Hug et al., 2016*; *Waite et al., 2020*). Due to these non-monophyletic properties, Pseudomonadota (Proteobacteria) classes will be treated as individual phyla for the rest of this study (*Hug et al., 2016*; *Waite et al., 2020*). Bacteroidota (Bacteroidetes), Acidithiobacilla and Epsilonproteobacteria formed slightly isolated groups away from the larger clusters, suggesting that these could become more defined clusters if their sample sizes were larger (Fig. 1). Alphaproteobacteria is the only cluster without a solved structure, implying the presence of phyla specific variation that has not been captured by current structures (Fig. 1). Therefore, it is unlikely that all the bacterial phyla are well represented by the available ribosome structures, given that the solved structures in the multiple phyla cluster group together, instead of being evenly distributed throughout the clusters (Fig. 1).

## Evaluation of current structures indicates that *Bacillus subtilis* is the most representative

To evaluate whether current ribosome structures from non-pathogens are sufficiently representative of the available sequenced bacterial ribosomes, phylogenetic distances were calculated from the summed distance matrix between 13 published solved structures (Table S2) and 1,385 bacterial ribosomal sequences available in this study (*Fox, 2010*; *Sohmen et al., 2015*; *Crowe-McAuliffe et al., 2021a*; *Jha et al., 2021*; *Hentschel et al., 2017*; *Morgan et al., 2020*; *Murphy et al., 2020*; *Yang et al., 2017*; *Halfon et al., 2019*; *Eyal et al., 2015*). The solved structure with the lowest recorded distance for each species is considered to be the most representative and assumes that species with similar primary sequences will form similar tertiary structures. As the minimum distance to the nearest solved structure increases, it becomes more likely that current structures are not suitably representative. These underrepresented species should be prioritised for future ribosome structural studies. Overall, *B. subtilis* was considered to be the most representative structure for 382 species, followed by *A. baumannii* for 264 species and 167 for *M. smegmatis* (Fig. 1). *D. radiodurans*, *T. thermophilus* and unsurprisingly the archaeon *H. marismortui* were the

**Table 1 The ten highest priority ribosome structures to solve, with priority given to pathogenic species.**

| Proposed target species | Phylum | Min distance to closest structure[1] | Avg min distance for phylum[2] |
|---|---|---|---|
| *Campylobacter jejuni* | Epsilonproteobacteria | 41.07 (*B. subtilis*) | 42.24 (10.11) |
| *Chlamydophila trachomatis* | Chlamydiota (Chlamydiae) | 40.25 (*B. subtilis*) | 39.58 (8.27) |
| *Opitutaceae bacterium* | Verrucomicrobiota (Verrucomicrobia) | 38.24 (*B. subtilis*) | 37.89 (16.00) |
| *Borrelia recurrentis* | Spirochaetota (Spirochaetes) | 38.00 (*B. subtilis*) | 39.41 (20.50) |
| *Singulisphaera acidiphila* | Planctomycetota (Planctomycetes) | 37.81 (*B. subtilis*) | 39.54 (21.48) |
| *Brucella melitensis* | Alphaproteobacteria | 37.73 (*A. baumannii*) | 38.78 (19.26) |
| *Chlorobium limicola* | Chlorobiota (Chlorobi) | 37.52 (*F. johnsoniae*) | 37.34 (5.64) |
| *Ureaplasma urealyticum* | Mycoplasmatota (Tenericutes) | 35.05 (*L. monocytogenes*) | 32.16 (26.65) |
| *Leptospirillum ferriphilum* | Nitrospirota (Nitrospirae) | 33.96 (*B. subtilis*) | 32.63 (16.34) |
| *Persephonella marina* | Aquificota (Aquificae) | 33.29 (*B. subtilis*) | 34.06 (14.81) |

Notes:
[1] Choices were based on the average minimum distances to a solved structure prior to the introduction of each proposed structure (See Table S1 for an extended list).
[2] The first value is the average minimum distance prior to the proposed structures being selected, and the value in brackets is the recalculated average distance after the proposed structures have been incorporated.

three least representative structures analysed and were not representative of any pathogenic species (Fig. 1), implying that the structures from these three species are becoming less representative with the introduction of ribosome structures from less divergent bacteria (*Fox, 2010*). *M. tuberculosis* was also representative for a small number of species, which is likely due to the presence of *M. smegmatis*, given that these two bacteria are closely related—effectively splitting the Actinomycetota (Actinobacteria) (Fig. 1).

This observation does not imply that one structure is necessarily sufficient to represent a complete phyla or class, with *E. coli* and *P. aeruginosa* representing 149 and 110 species respectively, accounting for most Gammaproteobacteria (Fig. 1). Whilst these structures are very similar, they do capture some important differences (*Halfon et al., 2019*). Each solved structure tends to be solely representative of the phyla it originated from, with *M. smegmatis* as the most representative of Actinomycetota (Actinobacteria), and both *E. coli* and *P. aeruginosa* representing Gammaproteobacteria exclusively. However, *B. subtilis* and *A. baumannii* were not representative of only their respective phyla, as the Gammaproteobacterium *A. baumannii* is representative of both Alpha- and Betaproteobacteria while the Bacillota (Firmicutes) *B. subtilis* was the most representative for the remaining phyla without a ribosome structure (Table 1). Therefore, we hypothesise that having at least one representative structure per phylum, or preferably one per class, would allow the majority of bacteria to be sufficiently represented.

Introducing new ribosome structures shows that an Epsilonproteobacteria representative, such as a *Campylobacter jejuni*, should be a high priority for further structural work. To simulate the effect of having at least one representative ribosome structure per phyla, we selected one proposed structure per phyla which had the smallest average phylogenetic distance to all other members in the respective phyla. The species that were initially considered to be potential representatives were selected based on how dissimilar they were to a species with a solved ribosome structure. These were selected by determining the phylogenetic distance between each isolate to a solved structure on the

combined rRNA and ribosomal protein phylogenetic tree, and selecting the structure with the shortest distance as the representative for that individual. Based on the distribution of these distances (Fig. S1), species with a distance to a solved structure greater than the lower quartile (11.41) were considered for the new ribosome structure analysis, as these species are most likely to be poorly represented by current structures. The available pathogens with a minimum distance above this threshold were prioritised due to their relevance in health research. The introduction of new proposed structures resulted in a decrease in the average minimum distance to a solved structure across each phylum, implying that having at least one structure per phylum is a suitable sampling strategy for improving representation (Table 1). Of the ten highest priority structures, only Epsilonproteobacteria, Chlamydiota (Chlamydiae) and Chlorobiota (Chlorobi) had an average minimum distance per phyla below the lower quartile threshold, after the proposed structures were introduced (Table 1). This suggests that the other phyla listed are more diverse and may require additional structures to capture the remaining variation, or that a representative non-pathogenic ribosome would be a more beneficial representative. *C. jejuni* was identified as the highest priority structure (Table 1), as it was the most representative Epsilonproteobacteria pathogen, had the largest average minimum distance observed across all phyla and is a WHO priority pathogen (*Tacconelli et al., 2018*). However, the selected ribosome for solving does not specifically need to be from a pathogenic strain, with either an attenuated lab strain (*Yang et al., 2017*; *Eyal et al., 2015*), or another species from the same genus (*Hentschel et al., 2017*; *Murphy et al., 2020*) being appropriate alternatives (Fig. 1).

## DISCUSSION

Our results reveal significant phylogenetic gaps in the existing cohort of solved ribosome structures. Filling these gaps will help identify clade-specific features of isolated bacterial clades that lack representative ribosome structures. This is important for the analysis of ribosome evolution, pathogen research, and potentially for antibiotic development.

There are two limitations with this approach to ribosome structure prioritisation: there is a bias towards species with available genomes, which are enriched in pathogenic species rather than being the most evolutionarily representative (*Wu et al., 2009*). In addition, our approach prioritises the most novel ribosomes within each major bacterial clade, with the risk of over-emphasising the most diverse ribosomes. This is somewhat mitigated by selecting the most closely related pathogenic species as representative targets.

While having at least one representative ribosome structure per phyla is expected to capture the bulk of structural variation (Table 1), there is no guarantee that primary sequence differences result in significant changes at the tertiary level. A prominent example of this is *H. marismortui* which, although it is an archaean ribosome structure, has been used as an alternative to bacterial counterparts (*Fox, 2010*), despite having no close relatives at the primary level (Fig. 1). Another consideration is that the structural variation observed may be species-specific, rather than shared across a phylum, with species-specific differences having been observed between *P. aeruginosa* and *A. baumannii*, yet no phyla-specific differences when compared to *E. coli* and other available structures (*Morgan*

*et al., 2020*; *Halfon et al., 2019*). However, these three structures were observed to be relatively divergent from each other based upon the MDS analysis (Fig. 1), which reinforces the *in vivo* observation that *P. aeruginosa* and *A. baumannii* ribosomes are more similar to each other than to *E. coli*, suggesting that phylogenetic analyses have the potential to reflect structural variation (*Morgan et al., 2020*).

Our list of high-priority bacterial ribosomes reveals that a number of bacterial clades are poorly represented by the existing solved structures (Table 1, Table S1). Working down this list gives an opportunity to identify the most variant ribosomes based upon sequence analysis, which in many instances will reflect diverse ribosome structures. These will allow the development of more specific classes of antibiotics, as well as provide further information regarding the specifics of ribosome structure evolution.

## ACKNOWLEDGEMENTS

The authors would like to thank Steven Gregory (University of Rhode Island) and Gerwald Jogl (Brown University) for their discussions and advice regarding this project.

### Funding

This work was supported by the Department of Biochemistry (University of Otago) as an Honours year research project and by an Australian Government Research Training Program (RTP) Scholarship. The funders had no role in study design, data collection and analysis, decision to publish, or preparation of the manuscript.

### Grant Disclosures

The following grant information was disclosed by the authors:
University of Otago.
Australian Government Research Training Program (RTP) Scholarship.

### Competing Interests

The authors declare that they have no competing interests.

### Author Contributions

- Helena B. Cooper conceived and designed the experiments, performed the experiments, analyzed the data, prepared figures and/or tables, authored or reviewed drafts of the article, and approved the final draft.
- Kurt L. Krause conceived and designed the experiments, prepared figures and/or tables, authored or reviewed drafts of the article, and approved the final draft.
- Paul P. Gardner conceived and designed the experiments, analyzed the data, prepared figures and/or tables, authored or reviewed drafts of the article, and approved the final draft.

### Data Availability

The raw data is available in the Supplemental Files. All data described was last accessed in September 2022. Custom scripts, details of parameters, and dependencies used, and the accessions for all downloaded data and the resulting curated alignments and trees are available on GitHub: https://github.com/helena-bethany/ribosomal-variation.

## Supplemental Information

Supplemental information for this article can be found online at http://dx.doi.org/10.7717/peerj.14969#supplemental-information.

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
