# Peer review of "Finding priority bacterial ribosomes for future structural and antimicrobial research based upon global RNA and protein sequence analysis"

_PeerJ, doi:10.7717/peerj.14969_

## Round 0.1 · original submission · Minor Revisions

I agree with the reviewers regarding their suggestions below so please address their comments in your revised manuscript.

Reviewer 1 ·

Basic reporting

The essence of this paper is that If we knew the ribosomal structure/sequences of a pathogen and a virtually identical non-pathogen it would be easy to explore the differences to understand what is responsible for pathogenesis. With this type of thing in mind the authors explore sequence space to determine for several pathogens what the closest non pathogen currently is. They are then able to indicate where in the existing gene set additional data would be the most useful. To accomplish they utilize current versions of various software tools that are widely used and up to date. They find that the distances are larger than one might expect in many cases. Overall, this article is well written and the supplementary materials are useful. An extremely minor issue --it might be useful to point out that using 3D structures in some manner would be problematic because one encounters different ribosome states etc. A possible addition to the supplementary materials would be a representative 16S or 23S secondary structure diagram illustrating where the sequence variation occurs for one of the pathogen/non-pathogen comparisons. One would like to get a sense of the extent to which the variation is clustered. Overall, this is a very well written paper that can be published as is.

Experimental design

Excellent.

Highly Original with well-defined questions. Paper detects a knowledge gap and how to explor how significant it is. State of the art methods are used and their purpose well explained.

Validity of the findings

Findings are Valid

Additional comments

Excellent paper

·

Basic reporting

The manuscript is clearly written and straightforward and self-contained with sufficient background and references. The raw data is shared on github.

Experimental design

Cooper et al. provide a succinct analysis and discussion of the
phylogenetic breadth of bacterial ribosomal RNA and protein
sequences. They evaluate the representativeness of current ribosome
structures in the context of all sequenced bacteria and known
pathogenic ones and identify phyla and pathogenic bacteria whose
ribosomes are relatively far from ribosomes with solved structures.
They make suggestions on which ribosome structures from pathogenic
bacteria should be solved next, and prioritize them based on reducing
as much as possible the average distance of existing ribosomes to
solved ribosome structures in each phyla. The methods, based on
alignment, tree building and multi-dimensional scaling, are clearly
explained.

Validity of the findings

The arguments regarding the prioritization of ribosomes are explained
well. The methods are clearly explained, and the necessary data to
reproduce their analysis is provided on github.

Additional comments

Major comment:

My understanding is that the entire alignment was used for
phylogenetic inference, including so-called insert columns in the
profile-based alignments. Nucleotides and amino acid residues in
insert columns are not 'aligned' by profile-based alignment methods
meaning that they should arguably be removed prior to phylogenetic
analysis. My guess is that this would not significantly affect the
results, but I am curious if the authors considered this approach, and
if so, if it did have any meaningful impact on the results. If they
did not, I would recommend rerunning part, if not all, of the
alignment->tree->MDS analysis with insert columns removed to determine
what impact it might have. If it does not have a meaningful impact, I
would be satisfied for the present analysis to be unchanged, but with
a comment added to the methods explaining that removing insert
columns was tested and had little impact with some supporting
evidence (a specific change in a distance between specific groups, for
example).
* * *
Minor comments:

Lines 71-73:
> This has led us to identify the highest priority groups of bacteria
> that would benefit most from further analysis of their ribosome
> structures, from both a medical and evolutionary perspective.

I suggest this sentence be reworded because as written it implies the
bacteria will benefit from further analysis of their ribosomes, which
I don't think is intended meaning.

--

Lines 80-81:
> Due to the limitations (recently renamed "Mycolicibacterium smegmatis")
> and requirements of X-ray crystallography, ...

The 'recently renamed' part is misplaced, and recurs on line 87.

Lines 161-163:
> Only ribosomal genes that are conserved across the majority of the
> species in our dataset were used. Therefore, ribosomal proteins uS2,
> uS4, uL15, uL23, uL24 and uL30 were removed as there were no
> homologous sequences within Epsilonproteobacteria.

Can you be more specific about what you mean by 'majority'? Do you
just mean > 50%? And if so, are the proteins listed the only ones not
present in > 50% of species in your dataset? And were they only
missing from Epsilonproteobacteria? I think a little more detail or
precise language here is warranted.

--
Figure 1:

I found myself rereading the legend and explanation in the text (lines
165-178) several times in an effort to grasp the information in this
figure. I have the following suggestions to make it easier to
understand:

1. Change either the species with a solved ribosome structure (N=13)
or the known pathogenetic bacteria without a structure (N=38) from
a solid black point to a black point outline (not filled) so the
reader can discriminate between the two. I realize the 13 are
labelled, but still think using different types of points is a good
idea.

2. For the Proteobacteria classes, which are each a different color,
make these points outlines (not filled circles), but still of
different colors, so that the reader understands all colored
outlined points are Proteobacteria, and all colored filled points
are non-Proteobacteria, and thus each filled color represents a
different phyla.

Also in lines 165-167:
> The resulting MDS plot shows four main clusters; two illustrate the
> non-monophyletic nature of Proteobacteria (38), one dominated by
> Actinobacteria and the fourth consisting of the remaining bacterial
> phyla (Fig. 1).

I recommend listing the classes that represent the two Proteobacteria
clusters by writing something like "(Alphaproteobacteria representing
one of these clusters or and Betaproteobacteria plus
Gammaproteobacteria representing the other)".

And after "remaining bacterial phyla", I recommend adding "and
Proteobacterial classes".

- At the end of the figure 1 legend, add the github url.

Reviewer 3 ·

Basic reporting

Cooper and colleagues apply sequence clustering to map out the diversity of bacterial (and archaeal) ribosomes and plot on the resulting map the locations of currently available structures. The motivation is two-fold: 1) to identify the gaps in our structural understanding of ribosomal diversity 2) provide the guide that would instruct the researchers as to which of the available structures is the best comparison for the ribosome from the species there are interested in.

I have several minor comments / suggestions.

1. Would it make sense to run the clustering for 50S / 30S separately and compare with the results for complete 70S? The authors bring up the specificity of antibiotic action as one of the motivations for their work, and antibiotics tend to work on a specific subunit. Therefore, if 30S and 50S cluster differently from complete 70S, it would mean that for 50S- and 30S-targeting antibiotics the most relevant existing structure might be different.

2. The authors use antibiotics as one of the arguments. Note that the antibiotics interact with a limited set of (mostly RNA-based) sites, while the clustering used by the authors uses the information that originates from complete rRNA and almost full set of r-proteins. It is possible that the metric / clustering is misleading, at least for some classes: e.g. maybe diversity of the PTC structure/sequence does not follow the overall clustering. I would at least outline this as a potential limitation.

3. I would provide the alternative (i.e. current) nomenclature as well (e.g. Firmicutes are now Bacillota). I am not a fan of it either, but it could improve the searchability / findability of the paper in the years to come.

4. In the introduction I would cite the recent Listeria monocytogenes high-res cryoEM structure https://pubmed.ncbi.nlm.nih.gov/36300626/

Experimental design

no comment

Validity of the findings

no comment

Additional comments

no comment

Reviewer 4 ·

Basic reporting

This manuscript explores the bacterial diversity of the ribosome (by a combined approach that jointly analyzes the similarity of rRNA and ribosomal proteins) from the perspective of identifying species targets for structural analysis that could help expand the repertoire of antibiotic approaches based on the ribosome. The manuscript proposes a list of 10 species that they recommend should be prioritized for structural characterization of their ribosomes.

This manuscript is a timely call for attention, specially with the increasingly common use of Cryo-em. I did not realize that half of the antibiotics target the ribosome. The method is also solid and straightforward. One comment resonates with me, that I think the authors could have explored some more, given in line 98. It suggests that even without need for newer crystal structures, more diverse antibiotics could be designed by focusing on the not so conserved regions of the ribosome.

Experimental design

The manuscript uses a not too common "global analysis" by combining the similarities observed in the rRNA and the ribosomal proteins. Could you provide an argument of what you expect to win from the combined comparison of rRNA with that of the ribosomal proteins? Could you anticipate an advantage of this method if applied to other RNA-protein complexes such as the spliceosome? (is the spliceosome an antibiotic target?)

Other comments that hopefully could improve the clarity of the presentation:

- The count of ribosomal proteins analyzed is not consistent within the paper. Line 133 mentions "32 universally conserved proteins", but later (line 160) "we combined 28 phylogenetic trees for each conserved rRNA and protein", which implies 26 ribosomal proteins?

- I recommend that the Method section gets edited with more breaks for better readability. For instance, the sentence "One representative sequence was retained per species for each rRNA and protein sequence," is used way before the reader realizes that the analysis involves ribosomal protein coding sequences, not all protein coding genes.

- Figure1: using black dots both for species with solved ribosome structures as well as known pathogenic bacteria without structure is confusing at best.
One needs to read the sentence after "Species with a solved ribosome structure (N=13), or known pathogenic bacteria without a structure (N=38) have been highlighted with a black dot.", to understand that those with a parenthesis and a number are the ones with solved structures versus not.

I recommend using different colors, and also to label the dots for epsilonbacteria and alphaproteobacteria and others that are now unlabeled. Using different dot colors would make that possible.


- Please, provide an introduction to "multidimensional scaling". What does it measure? How does it measure it? Why was it considered a good metric?

- Please, explain "minimum phylogenetic distance", how is it calculated?

- Related to the previous, minimum distance "greater than 12.86", please explain rational for that number. Maybe show the (cumulative) histogram of distances for all species?

Validity of the findings

no comment

---

## Round 0.2 · accepted · Accept

Thank you for addressing the reviewers' comments so thoroughly. I have assessed these myself and am happy with the current version.